# Improving Generalization of Norwegian ASR with Limited Linguistic Resources

**Per Erik Solberg**
National Library of Norway
Oslo, Norway
`per.solberg@nb.no`

**Pablo Ortiz**
Telenor Research
Oslo, Norway
`pablo.ortiz@telenor.com`

**Phoebe Parsons**[1]    **Torbjørn Svendsen**[1]    **Giampiero Salvi**[1,2]
1) Department of Electronic Systems, NTNU, Trondheim, Norway
2) KTH Royal Institute of Technology, EECS, Stockholm, Sweden
{`phoebe.parsons,torbjorn.svendsen,giampiero.salvi`}`@ntnu.no`

## Abstract

With large amounts of training data, it is possible to train ASR models that generalize well across speakers and domains. But how do you train robust models when there is a limited amount of available training data? In the experiments reported here, we fine-tuned a pre-trained wav2vec2 ASR model on two transcribed, Norwegian speech datasets, one with parliamentary speech and one with radio recordings, as well as on combinations of the two datasets. We subsequently tested these models on different test sets with planned and unplanned speech and with speakers of various dialects. Our results show that models trained on combinations of the two datasets generalize better to new data than the single-dataset models, even when the length of the training data is the same. Our lexical analysis sheds light on the type of mistakes made by the models and on the importance of consistent standardization when training combined models of this kind.

## 1 Introduction

Automatic speech recognition has experienced tremendous development in the past decades. New models have improved in sophistication as well as complexity, see e.g., (Chorowski et al., 2015; Chan et al., 2015; Amodei et al., 2016; Synnaeve et al., 2019; J. Li et al., 2020; Baevski et al., 2020; B. Li et al., 2021; Radford et al., 2022) and references therein. However, these models are still at best as good as the data they are trained on. Spoken language presents many dimensions, such as degree of spontaneity, dialectal variation, task domain, and age of speaker, that affect the performance of a speech recognizer. Models trained on a specific combination of those factors will perform poorly in other conditions.

With sufficient computing capacity and data, this problem can be mitigated by training large models on even larger sets of data spanning several conditions, both in terms of the spoken content as well as transcription standards (e.g. Whisper, Radford et al., 2022). However, when computing power and quantity of data are limited, it is necessary to provide a dataset with a common transcription standard and enough variety of acoustic and linguistic features to fully encapsulate the target language. Such a dataset can then be used to fine-tune a large pre-trained model such as wav2vec2 (Baevski et al., 2020); a process less computationally expensive than training from scratch.

Until recently, freely available speech datasets suitable for training Norwegian ASR models only contained manuscript-read speech. However, in 2021, the Norwegian Language Bank released the Norwegian Parliamentary Speech Corpus (NPSC), containing 126 hours of transcribed speech from Stortinget, the Norwegian Parliament. The dataset contains a decent amount of spontaneous speech, as well as dialectal variation. Solberg and Ortiz (2022) showed that ASR models trained on the NPSC were better at transcribing spontaneous speech and dialects than models trained on manuscript-read speech only. Researchers at the National Library of Norway have fine-tuned wav2vec2 ASR models on the NPSC training split. Their 1B parameter model[1] obtained a word error

---

[1] `https://huggingface.co/NbAiLab/nb-wav2vec2-1b-bokmaal`

rate (WER) of 6.4% on the NPSC test split (De la Rosa et al., 2023).[2]

However, when we apply ASR systems trained on the NPSC to speech data such as broadcasts and informal speech, they tend to perform less well than on parliamentary speech. This is understandable since parliamentary speech is rather formal and contains few interruptions and instances of real conversation. Moreover, the topics discussed in Parliament are relatively restricted, and words which occur frequently in everyday conversations may be rare or nonexistent in such a dataset. Finally, the recording conditions are very homogeneous.

In this paper, we investigate the effect of finetuning wav2vec2 models on more varied Norwegian training data. Two models are trained on single training sets with unplanned speech, the NPSC and the broadcasting dataset Rundkast, while two others are trained on combinations of these datasets. We show that the models trained on combined data sources generalize better to new data than the models trained on single datasets and that this effect cannot be attributed to the length of the training data. Our analyses also show that *standardization*, the process of making transcriptions as uniform as possible across datasets, is key when combining training data in this way.

The outline of the rest of the paper is as follows. In Section 2, we describe the datasets used in the experiments. Section 3 reports on how the models were trained, as well as the experimental setup. The results of the experiments are described and discussed in Section 4, and we also use a technique from corpus linguistics called *keyword analysis* to get a deeper understanding of the differences between the models. Section 5 concludes the paper and suggests some avenues for further development and research, based on our results and the state of Norwegian ASR.[3]

## 2 Datasets for Norwegian ASR

There are a number of datasets that can be used for training and testing Norwegian ASR systems. This section describes the most important, freely accessible Norwegian speech datasets, as well as one which is not freely accessible, Rundkast. All of these datasets have transcriptions in Norwegian Bokmål, the most commonly used of the two written standards of Norwegian, while some also have transcriptions in Nynorsk. In the experiments of this paper, we only use the Bokmål transcriptions.

### 2.1 NST

The now defunct company Nordisk språkteknologi ('Nordic Language Technology'; NST) made a large dataset for training and testing their ASR system in the late 90s and the beginning of the millennium. This dataset has been shared with an open license at the Norwegian Language Bank at the National Library of Norway since 2011.[4] The NST dataset includes around 540 hours of manuscript-read speech in Bokmål from close to 1000 informants. They read sentences from newspapers, but also some repeated words and sequences of numbers. Since the dataset only contains planned speech, there are few instances of hesitations, repetitions, false starts, etc., which are more frequent in unplanned speech. The speakers come from different dialect areas, but since the speech is scripted, the speech deviates less from the Bokmål norm than unscripted speech. This accounts for why models trained only on the NST generalize less well to different dialects than systems trained on both NST and the NPSC in (Solberg and Ortiz, 2022).

### 2.2 NPSC

The NPSC was developed by the Norwegian Language Bank in 2019-2021 as an open dataset for ASR of Norwegian unscripted speech (Solberg and Ortiz, 2022).[5] This dataset consists of about 126 hours[6] of recordings of meetings from 2017 and 2018 at the Norwegian Parliament. These are transcribed manually by trained linguists. There are transcriptions both in Bokmål (87%) and Nynorsk (13%). Individual speakers are transcribed consistently in one or the other written standard, following the practice in the official parliamentary proceedings. There are different versions of the transcriptions intended for different use cases (cf. Solberg and Ortiz, 2022,

---

[2]The data processing and training setup for those models are somewhat different from the NPSC-trained model in the experiments reported in this paper, and thus the results slightly differ.

[3]Code for the experiments is available at `https://github.com/scribe-project/nodalida_2023_combined_training`.

[4]`https://www.nb.no/sprakbanken/en/resource-catalogue/oai-nb-no-sbr-54/`

[5]`https://www.nb.no/sprakbanken/en/resource-catalogue/oai-nb-no-sbr-58/`

[6]Excluding breaks. The total duration is 140 hours.

sect. 2.2). We use the version in which numbers, dates and years are written with letters instead of digits, and abbreviations are not used. The NPSC also contains metadata about non-standard and/or dialectal words, which we use to standardize transcriptions, as described in subsection 2.5. There are 267 speakers in the dataset.

## 2.3 NB Tale

NB Tale is also an open speech dataset from the Norwegian Language Bank, published in 2015.[7] It is divided into three modules, two of which are used in this paper: Module 1 consists of manuscript-read sentences from newspapers by native speakers of different Norwegian dialects. The sentences are chosen to cover as many phonological phenomena as possible and are transcribed both orthographically and phonetically. Only the orthographic transcriptions are used in the experiments reported here. Some of the sentences in the dataset are read by all speakers, while others are read by a subset of the speakers or only one speaker. There are detailed metadata about each speaker, including dialect, age, and gender. Module 3 consists of recordings of the same speakers as module 1, as well as some non-native speakers (excluded from our analyses), speaking freely for 2 minutes on a subject of their choice. These are orthographically transcribed. There are 380 speakers in NB Tale. Module 3 is Bokmål only. 14.2 % of module 1 is in Nynorsk.

## 2.4 Rundkast

Rundkast, the only one of these datasets which does not have an open license, was developed by the Norwegian University of Science and Technology in 2005-2006 (Amdal et al., 2008). It consists of 77 hours of orthographically transcribed radio news and actuality shows from NRK, the Norwegian state broadcaster.[8] The written standard of the transcriptions is either Bokmål (80%) or Nynorsk (12%), depending on the dialect of the speaker.[9] Only the Bokmål transcriptions are used here. The dataset includes read news, interviews, debates, and commentary.[10]

---

[7]https://www.nb.no/sprakbanken/en/
resource-catalogue/oai-nb-no-sbr-31/

[8]A small subset is also phonetically transcribed.

[9]8% are tagged as neither.

[10]Due to inconsistent uses of speaker names in Rundkast, it is not possible to make a reliable speaker count for this dataset.

## 2.5 Standardization and usage of combined data

Naturally, the four datasets described above have different transcription standards and metadata. We provide a set of standardizing procedures that aim to unify transcriptions from the aforementioned corpora such that they can be combined and used together consistently.[11] Of particular importance is the treatment of digits, abbreviations, non-verbal noises (e.g., hesitations), and non-standard and dialectal words across datasets. The most important changes to the original transcriptions are:

- Remove all special characters and punctuation except for "é" and "-", which appear often in Norwegian and can make a difference in the words' meaning.

- Substitute all the digits by letters according to how they are pronounced. While numbers, years and dates are written with letters in the datasets used here, the original transcriptions of the NPSC include some company names etc. which contain digits.

- Substitute non-verbal noises by three variants: "mmm" (nasal hesitation), "eee" (vowel hesitation), and "qqq" (other non-language vocal sounds such as laughter or coughing).

- Non-standard words and dialectal words are by default not standardized, for the purpose of having orthographic transcriptions that reflect as close as possible what is actually said. As a consequence, the transcriptions may contain words that are not in standard dictionaries of Bokmål.

The train, test and validation splits are performed on NST, NPSC and Rundkast individually, while NB Tale modules 1 and 3 are used for testing purposes only. For NPSC, we use the official splits. Parliamentary meetings are used as the minimum unit, i.e. they are not divided across different splits. This is to minimize the overlap in topics and vocabulary across splits (Solberg and Ortiz, 2022). For the NST, there was only an official train and test split. We, therefore, split the official test set into a test and validation set randomly. For Rundkast we performed a split using full programs

---

[11]The code that implements all the procedures described in this section, as well as the procedures for creating data splits, is available at https://github.com/scribe-project/asr-standardized-combined

as the minimum unit, and allocate programs with the largest number of different speakers to the test set, and then to the validation set. This is to better evaluate the generalization capabilities of the models. The Rundkast splits were kept as close as possible to the proportion 80:10:10 in terms of duration of the train, test and validation sets, respectively. These are also the proportions of the official NPSC splits (Solberg and Ortiz, 2022, sect. 3.4).

## 3 Experiments

The goal of our experiments is to verify to what extent we get improvements in performance when training on more varied spontaneous data than the NPSC data alone. To this aim we fine-tuned wav2vec2 models on both Rundkast and the NPSC individually, as well as on combinations of the two datasets. We then test the models on test sets from the same domain as the training set, as well as from different domains using the NB Tale and the NST corpora.

### 3.1 ASR framework and hyperparameters

All models described in Section 3.2 are based on the wav2vec2 architecture (Baevski et al., 2020). Inspired by the fine-tuning of Norwegian 300M parameter models in (De la Rosa et al., 2023), we used the Swedish 300M parameter wav2vec2 model trained in Sweden by Kungliga Biblioteket[12] (Malmsten, Haffenden, and Börjeson, 2022) as a starting point, and fine-tuned it to different sets of data. All training sessions used the default hyper-parameters in Huggingface transformer implementation, with the exception of the initial learning rate for the Adam optimizer that was set to $10^{-4}$. All models were trained for 30 epochs, and the checkpoint with the lowest WER on the validation set was chosen for the recognition experiments on the test set.

Recognition was performed in two different settings for each model (without and with language model). In the first, the most likely token for each time step is first computed based on the output activations of the model. The sequence of best tokens is then passed to the tokenizer for decoding into words. In the second setting, the output activations of the model are passed directly to the tokenizer that uses beam search and a language model to produce the textual output. In this set-

ting, we used the 5-gram model produced by researchers at the National Library of Norway[13].

As a reference we also performed recognition on our test sets with the large Whisper model (1.55 billion parameters) trained on a total of 680000 hours of (multilingual) speech, including 266 hours of Norwegian. In this case, the model is used without fine-tuning. When computing word error rates, in this case, we used Whisper's 'basic' text normalizer, followed by normalization of most numerals to minimize the discrepancies between reference text and the Whisper transcriptions. However, the corresponding results are not directly comparable with the wav2vec2 results because Whisper is trained to produce a loose transcription of speech rather than word-by-word transcriptions. Those results will therefore only be used for discussion. It is worth noting that both wav2vec2 and Whisper can output any sequence of characters (not only words out of a fixed vocabulary). For this reason, it is interesting to analyze not only the number of word errors, but also spelling variants or mistakes. This will be done in details in Section 4.3.

### 3.2 Models

We fine-tuned four models on four different training sets.[14] To distinguish models from training sets, model names are written in small caps. The model trained on the NPSC only is called STORTINGET, the name of the Norwegian Parliament.

**RADIO** The RADIO model is trained on Bokmål segments from the Rundkast training set with a segment length above 1 second and below 15 seconds. This amounts to 43.6 hours of audio.

**STORTINGET** The STORTINGET model is trained on Bokmål segments from the NPSC training set with a segment length above 1 second and below 15 seconds, which adds up to 70.3 hours of audio. That is, 80.4% of the original Bokmål training set.

**COMBINED SHORT** The COMBINED SHORT model is trained on a random sample of segments from the training sets of RADIO

---

[12] https://huggingface.co/KBLab/wav2vec2-large-voxrex

[13] https://huggingface.co/NbAiLab/nb-wav2vec2-kenlm

[14] These models are published on Huggingface: https://huggingface.co/scribe-project.

and STORTINGET (half of the total duration comes from each dataset). The total duration of the training data for COMBINED SHORT is 70.4 hours, only 4 minutes longer than the training set of the STORTINGET model. Thus, we cannot attribute performance differences between the two models to the size of training data.

**COMBINED LONG** Finally, the COMBINED LONG model is trained on the combination of the training sets of RADIO and STORTINGET without leaving anything out, which amounts to 114 hours.

### 3.3 Test Sets

The models were tested on NB Tale modules 1 and 3 and the test sets of the NPSC, Rundkast and NST. In all these datasets, we have filtered out segments shorter than one second and longer than 15 seconds.

Table 1 gives an overview of the duration, domain and use of the dataset samples used in the experiments.

## 4 Results and Analyses

### 4.1 Results per dataset

Table 2 reports the WER per dataset, for each model. When looking at the results, it makes sense to inspect the NPSC and Rundkast test sets separately from the others, since those are the test sets of the models' training data. As expected, the RADIO model outperforms the STORTINGET model on the Rundkast test set (17.8% vs. 24.0%), and conversely, the STORTINGET model outperforms the RADIO model on the NPSC test set (9.3% vs. 19.5%). The COMBINED SHORT model has a slightly better WER than the RADIO model (17.2%) on the Rundkast test set. This is not necessarily surprising, given that the COMBINED SHORT model is trained on a larger dataset than the RADIO model. Still, COMBINED SHORT is trained on less Rundkast data than RADIO, and adding data from a different domain seems to have a positive effect. The COMBINED LONG model gives the best score on the Rundkast test set (15.9%). For the NPSC test set, COMBINED SHORT does not improve on STORTINGET (9.9% vs. 9.3%). Again, COMBINED LONG has the lowest score (7.9%).

COMBINED LONG is the best model on the datasets which are not the models' test sets: NB

Tale modules 1 and 3 and NST test. Furthermore, COMBINED SHORT has a better WER than both non-combined models. Since COMBINED SHORT is trained on the same amount of data as STORTINGET, this effect cannot be attributed to the size of the training set, so the improved generalization of the model seems to be due to the mixing of datasets. It is interesting to observe that RADIO performs better than STORTINGET on NB Tale 1 (24.5% vs. 25.8%) and on NST (10.6% vs. 11.2%). RADIO performs worse than STORTINGET on NB Tale 3, however (28.9% vs. 25.8%). NB Tale 1 and NST are manuscript-read datasets, while NB Tale 3 is spontaneous, which may be part of the explanation for the difference. It is not clear why, though, as neither STORTINGET nor RADIO are primarily manuscript-read.

When we look at the results with a language model, we see the same general trends, but with lower WER values: COMBINED SHORT is consistently better than STORTINGET on all datasets, with the exception of the NPSC test.

In addition to the limited resource models, we also tested the 1.55 billion parameter Whisper model (Radford et al., 2022) which is trained on massive amounts of heterogeneous data, from different sources, task domains and in multiple languages. As previously mentioned, the results from Whisper are not directly comparable, because the style of transcriptions from this model is different. However, the Whisper performance on NB Tale 1 and 3 and on NST is as good or better than the fine-tuned models (17.5%, 22.6% and 9.5% WER respectively). On the other hand, Whisper's performance on NPSC, Rundkast is clearly lower than our fine-tuned models (16.6% and 25.0% WER respectively). A possible explanation is that data from the NB Tale and NST datasets is not included in the training material for fine-tuning, reducing the effect of fine-tuning for these test sets, while Whisper benefits from the much larger and heterogenous training data.

### 4.2 Results per dialect

We report results per dialect in Table 3. Here we focus on NB Tale module 3 only because this part of the dataset contains spontaneous speech by a balanced set of dialect speakers, and is therefore well-suited for dialect-wise testing.

The dialect region *east* includes the Oslo re-

| Dataset | train | test | validation | Domain | Use |
|---|---|---|---|---|---|
| NPSC | 70.3h | 9.1h | 9.6h | mixed | train/test/validation |
| Rundkast | 43.6h | 5.9h | 5.5h | mixed | train/test/validation |
| NST | n/a | 25.6h | n/a | read | test |
| NB Tale 1 | n/a | 9.3h | n/a | read | test |
| NB Tale 3 | n/a | 7.4h | n/a | spontaneous | test |
| Combined short | 70.4h | n/a | 9.7h | mixed | train/validation |
| Combined long | 114.0h | n/a | 15.1h | mixed | train/validation |

Table 1: Overview for the datasets samples used in the experiments.

| Model | (training hours) | NPSC | Rundkast | NST | NBT1 | NBT3 |
|---|---|---|---|---|---|---|
| RADIO | (43.6h) | 19.5 (14.9) | 17.8 (15.3) | 10.6 (7.1) | 24.5 (19.5) | 28.9 (23.2) |
| STORTINGET | (70.3h) | 9.3 (8.1) | 24.0 (20.5) | 11.2 (7.8) | 25.8 (20.7) | 25.8 (21.5) |
| COMBINED SHORT | (70.4h) | 9.9 (8.3) | 17.2 (14.6) | 9.2 (6.3) | 23.5 (19.1) | 24.4 (19.9) |
| COMBINED LONG | (114.0h) | **7.9 (7.1)** | **15.9 (14.0)** | **8.6 (6.0)** | **21.9 (18.0)** | **23.0 (19.2)** |

Table 2: Word error rates (%) per model and test set. The best results with limited linguistic resources (wav2vec2) are shown in **bold** and the second best are underlined. Results in parentheses are obtained by combining the wav2vec2 models with a 5-gram language model.

gion and the counties in the southeastern part of Norway. *South* groups together the dialects in the county of Agder on the south coast. *West* includes the dialects on the southwest coast of the country in the counties of Rogaland, Vestland, and southern Møre og Romsdal. *Mid* includes the dialects in the county of Trøndelag and northern Møre og Romsdal, while *north* covers the dialects north of Trøndelag, in the counties of Nordland and Troms og Finnmark.

As we saw in the previous section, RADIO has poorer performance than STORTINGET on NB Tale module 3 globally, and we see that this difference is reflected in all dialect regions.

All models perform best on the eastern dialects. This is not surprising, as more than half of the population lives in this region (Thorsnæs, 2023) and there is a bias in the models' training data towards the dialects in this region. Moreover, many of the eastern dialects, in particular those in the Oslo region, are close to the written Bokmål norm.

From Table 3 we can see that all models struggle most with the *mid* and *west* dialects. Many dialects from these areas have inflections and lexical forms of words which differ substantially from Bokmål. Moreover, the models are exposed to limited amounts of western Norwegian, as many of the speakers from the west coast are transcribed in Nynorsk in Rundkast and the NPSC. Nynorsk

transcriptions are filtered out in the datasets used for training and testing these models.

COMBINED SHORT improves on the single dataset models for all dialect regions. The improvement from RADIO to COMBINED SHORT is substantial across regions, ranging from a relative improvement of 13.5% for *east* to 17.9% for *south*. Again, this is not surprising, as COMBINED SHORT is trained on more data. From STORTINGET to COMBINED SHORT, there are also improvements, although less substantial: COMBINED SHORT improves on STORTINGET by a relative 10.2% for *east*, while for the *mid* region, the WER is almost the same for the two models. For the other regions, the relative improvements are below 7%.

### 4.3 Lexical analysis

To better understand the kinds of errors the models make we have used a technique from corpus linguistics called *keyword analysis* (Dunning, 1994; Pojanapunya and Todd, 2018, and references therein). Keywords are words that have a surprising frequency, either surprisingly high or surprisingly low, in a target corpus relative to a reference corpus. Words are assigned a value indicating their keyness. Two common statistics used to compute keyness are log-likelihood (LL) and $\chi^2$. In this study, we will use LL, which is the more reliable statistics when the expected frequency of

| Model | east | west | mid | north | south |
|---|---|---|---|---|---|
| RADIO | 22.3 | 32.7 | 32.0 | 27.1 | 25.7 |
| STORTINGET | 21.5 | 28.6 | 27.7 | 24.6 | 22.3 |
| COMBINED SHORT | 19.3 | 27.2 | 27.4 | 22.9 | 21.1 |
| COMBINED LONG | **18.2** | **25.8** | **25.5** | **21.5** | **20.5** |

Table 3: Word error rates (%) per dialect in the NB Tale module 3 test set. The best results are shown in **bold** and the second best are underlined.

a word is low (Dunning, 1994).

Keyword analysis is often used to characterize a text's genre or identify its ideological underpinnings. It can also be used to generate term lists for a given field or topic (Pojanapunya and Todd, 2018). In this study, we used keyword analysis to identify word forms that characterize a machine transcription relative to the ground truth. For each machine transcription, we looked through the list of the 100 words with the highest LL value. Such a keyword list can reveal the words contributing to the machine transcription WER. Words that have either an unusually high or an unusually low relative frequency in the machine transcription relative to the ground truth, will get a high LL and will therefore appear in the keyword lists. Both cases may reveal properties of the transcription. One limitation of this method is that word forms occurring only once in the target corpus and never in the reference corpus will get a low LL and not appear in the keyword list. Therefore, we may miss misspellings.

There are many instances of incorrectly spelled words with a high frequency in the automatic transcription and a zero frequency in the ground truth. Often, the correctly spelled version is also present in the list, with a higher frequency in the ground truth than in the automatic transcription. Typically, these words have a spelling that is surprising given the pronunciation of the word, such as foreign company names and loan words. The genitive of Apple, "apples" has a frequency of 241 in the ground truth and 0 in the STORTINGET transcription.[15] The STORTINGET list contains misspellings of this name, only occurring in the automatic transcription, such as "appels" and "apels". Similarly, "rock" occurs 37

times in the ground truth, but never in the ASR output from STORTINGET.

We see a similar phenomenon with uncommon words. As mentioned, the sentences in NB Tale module 1 are chosen to cover as many phonological phenomena as possible, and many sentences are repeated by several or all informants. As a consequence, there are quite a few uncommon words in that dataset, and some of them appear in the keyword analysis. An example is the word "stokkmaur" ('Carpenter ant'), which occurs 240 times in the ground truth, but only 56 times in the STORTINGET transcriptions. The STORTINGET list contains two misspellings of this word, however: "stokmaur", "stokkmør". RADIO prefers to spell this word "stockmaur".

There are quite a few examples of words where a vowel is left out, possibly due to fast speech, such as "tittlen", instead of "tittelen" ('the title'). However, there are not many obvious examples of dialect pronunciations, except for some very frequent function words. This does not necessarily mean that dialectal pronunciations do not contribute to the WER of the models: Dialectal transcriptions could be hapaces, forms occurring only once, which will not get a high LL value. The ground truth transcriptions of native speakers has 9672 hapaces, while hapax count for STORTINGET and RADIO is more than twice as high. They have 20591 and 20215 hapaces respectively. The hapax count goes down to 18676 for COMBINED SHORT and 17613 for COMBINED LONG. An inspection of a sample of the hapaxes from the different models which don't also occur in the ground truth, reveal that they are, to a large extent, misspellings. It is, however, hard to tell from reading the misspellings whether they are of dialectal origin or not. The hapax count goes further down when a language model is used (STORTINGET: 13576, RUNDKAST: 13125, COMBINED SHORT: 12076, COMBINED LONG: 12065), which indicates that the language model

---

[15]All words are spelled with lowercase letters in the datasets. Both NB Tale module 1 and NST contain many repeated sentences, which likely accounts for the high frequency of this genitive form. Note also that genitive forms are spelled without an apostrophe in Norwegian.

| Model | NPSC | Rundkast | NST | NBT1 | NBT3 |
|---|---|---|---|---|---|
| RADIO | 17.4 (13.8) | 16.0 (13.2) | 9.7 (7.1) | 22.5 (18.5) | 27.2 (21.4) |
| STORTINGET | 8.8 (7.4) | 21.1 (17.0) | 11.1 (7.8) | 25.3 (19.9) | 24.4 (19.5) |
| COMBINED SHORT | 9.3 (7.6) | 15.4 (12.5) | 8.7 (6.3) | 21.7 (18.1) | 22.8 (18.2) |
| COMBINED LONG | **7.2 (6.2)** | **14.1 (11.9)** | **7.8 (6.0)** | **19.5 (17.0)** | **21.2 (17.5)** |

Table 4: Word error rates (%) per model and test set with hesitations removed and with standardization of compounds and acronyms. Best results are in **bold** and second best are underlined. Results in parenthesis are obtained combining the wav2vec2 model with a 5-gram language model.

reduces the number of spelling mistakes.

The test sets have special markings for hesitations, and the models are trained to produce such markings too. This appears to be a source of errors. In particular, nasal hesitations, marked as "mmm", occur 994 times in the ground truth, but almost never in the automatic transcriptions, and it is the word with the highest LL in the keyword analyses of all the models. This kind of error does not affect the semantics of the transcription, and markings of hesitations will presumably be removed in many downstream applications.

Another source of errors which does not impede the understanding of the transcriptions, is insufficient standardization of the different datasets. When comparing the analyses of the two single-dataset models, it turns out that STORTINGET tends to transcribe compounds without a hyphen, e.g. "arbeiderpartipolitikeren" ('the Labor Party politician'), while RADIO tends to use a hyphen: "arbeiderparti-politikeren". The datasets STORTINGET is trained on, the NPSC, also transcribes compounds without a hyphen while Runkast, which RADIO is trained on, uses a hyphen, and this difference is not captured by the standardization routines described in section 2.5. There is a similar issue with acronyms. STORTINGET transcribes acronyms such as NRK, the national broadcaster, as "nrk", while RADIO separates each letter with a space, "n r k", also due to a difference in the training data which is not captured by the standardization routines. Unsurprisingly, the combined models produce a mix of these standards.

While hesitations and differences in transcription standards contribute to the WER, they are in a sense less important than misspellings and wrong words, which may affect the comprehension and the usability of the transcriptions. We would, therefore, like to check to what extent these errors contribute to the WER. Can the higher performance of COMBINED SHORT compared to the single-dataset models be explained entirely by these errors? To check this we have made a version of the ground truth and the automatic transcriptions where hesitations are removed, compounds are written without hyphens, and where a number of the most frequent acronyms are written without a space between them. Table 4 reports the results across datasets with these standardizations. The values in parentheses are the WER with a language model. The results should be compared to those in table 2. As before, the COMBINED SHORT model outperforms the single dataset models on all datasets except the NPSC test set, where the STORTINGET model is better. To see the effect of this cleaning of hesitations, hyphens and spaces, we can look at the global WER across all the datasets (excluding foreign speakers). Before cleaning the global WER is 16.8% for STORTINGET, 18.1% for RADIO, and 14.7% for COMBINED SHORT without a language model. After cleaning, the global WERs are 16.0%, 16.5%, and 13.6% respectively. The relative improvement of the global WER from STORTINGET to COMBINED SHORT is 12.5% before cleaning and 15.0% after cleaning. For RADIO, the improvement is 18.8% before cleaning and 17.6% after cleaning. In other words, when we exclude the errors we have observed which stem from hesitation annotations or differences in transcription standard, the gap between STORTINGET and COMBINED SHORT becomes somewhat larger and the gap between RADIO and COMBINED SHORT becomes somewhat smaller, but COMBINED SHORT still improves on the single-dataset models. The difference between the single dataset models and the combined dataset models cannot be explained solely by the transcription of hesitations and the differences in transcription standards.

## 5 Conclusion and future work

In this paper, we have shown that training ASR models on a combination of parliament speech data from the NPSC and broadcast data from Rundkast results in better WER across different test sets than models trained on these datasets individually. This effect persisted even when we control for dataset length. While STORTINGET is slightly better on the NPSC test set than the combined model of similar length, COMBINED SHORT outperforms both single dataset models on all other test sets. In other words, the combined models generalize better to out-of-domain speech data, which makes them more suitable for downstream transcription tasks where different kinds of speech data may be encountered, such as meeting transcriptions and subtitling.

The study also highlights that it is important to standardize the training and test data when combining datasets in this way. This standardization may require an intimate knowledge of the transcription guidelines of the different datasets. Even though we had standardized the datasets prior to training, as described in section 2.5, we did not discover the differences in the treatment of acronyms and compounds until we investigated the ASR outputs in detail.

To be able to train on combinations of datasets, one needs to have access to ASR dataset of different types and genres. Before the release of the NPSC in 2021, there were no large, open datasets for ASR training with Norwegian unplanned speech. The NPSC may be released openly because parliamentary recordings are in the public domain. Due to copyright and privacy issues, it is more difficult to make a dataset with broadcast data such as Rundkast freely available. A recent report from the Norwegian Board of Technology points out that there are not enough open ASR datasets for Norwegian, and the datasets that exist are not sufficiently varied. It suggests different ways to increase the amount of open speech data, such as a major crowdsourcing initiative (Tennøe and Wettre, 2022). While we wait for more open data, it may be possible to train ASR models on a combination of open and non-open datasets and release the resulting models openly.

Finally, the results we obtained with Whisper are not comparable to ours using WER. This is because the model is trained to produce transcriptions of different standards. This emphasizes the importance of developing new metrics that assess the semantic content of the transcriptions rather than the word accuracy.

## Acknowledgments

This work has been partially supported by the Research Council of Norway through the IKTPLUSS grant for the SCRIBE project[16] (KSP21PD).

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
