# OpenReview forum: "Improving Generalization of Norwegian ASR with Limited Linguistic Resources"
_NoDaLiDa/2023/Conference — NoDaLiDa 2023_

### Official Review · Reviewer_rpDk · 2023-03-08
**Review for paper92**

**Rating:** 7
**Confidence:** 5

**Review:**

This paper describes improving generalization of Norwegian ASR by fine-tuning pre-trained Wav2Vec 2.0 model on two datasets. The results show that models trained on combinations of the two datasets generalize better to new data than the single-dataset models, when the length of the data and transcription standartization are controlled. Additionally, lexical analysis provides insight on the type of mistakes made by the models and on the importance of consistent standardization of training data.

The paper is well written and easy to read, except maybe for Section 4.3 which is long and contains multiple various experiments and observations. Perhaps, it could be split. The method as applied in the paper is described well. The architecture, optimization, data and challenges are clearly described.

Strengths&weaknesses:

[+] Evaluation is performed on multiple different data sets covering multiple domains and dialects.
[+] Controls performed to make sure that improvements come from improved generalization of the model and no other factors.
[+] Great effort on lexical analysis, that provided interesting insights and allowed to discover the differences in the treatment of acronyms and compounds in different data sets.
[-] Fine-tuning of wav2vec 2.0 is not really novel these days.
[-] It's well-known that better results can be obtained if transcripts of combined datasets are uniform.


**Paper Type:**

Long paper

---

### Official Review · Reviewer_fezX · 2023-03-10
**Relevant ASR experiments, but limited novelty**

**Rating:** 5
**Confidence:** 4

**Review:**

Pros:
- Authors study and compare models on several in-domain and out-of-domain test sets, of which all but one are public.
- The paper includes a rare look at dialect-level performance and an analysis of the result.
- Authors study the errors behind the WER values using lexical analysis which gives nice insight to the errors.

Cons:
- Dataset details are reported inconsistently, a comprehensive table would be very helpful.
- The choice of wav2vec model should be justified and explained (currently there is none).
- Some of the hypotheses in the analysis section sound like authors are just guessing.

Detailed feedback:
 - For NPSC, how are the different versions of transcripts produced? Are they created by human transcribers or by a machine? You also mention that each have a different use case, could you provide a concrete example here?
 - Dataset details are reported very inconsistently and it is hard to get a comprehensive picture of your data. Some datasets miss speaker count, others miss the total hours, year of collection/publication or proportion of bokmål audio. It would be a big improvement if you had a table where you list e.g. total audio duration, portion of the data you actually used (hours of bokmål data), total speaker count (and speaker count in your experiments since you exclude data), domain (read, spontaneus, other). Addendum: I see that section 3.2 has some of this information. I think keyfigures from sections 2 and 3.2 should be combined to one table and the text should refer to that table for the details, instead of having the numbers dispersed throughout the article.
 - Is standardization applied to all of the datasets? NPSC already had digits written with letters and no abbreviations, so does it require standardization? If it doesn't, is the aim of the standardization to make other datasets resemble NPSC?
 - You write that you use the official split for NPSC, but then in the last sentence of the section you imply you created the split yourselves ("The NPSC and Rundkast splits were kept as close as possible to the proportion 80:10:10")?
 - Will you publish the splits you created somehow (at least for the public datasets), so others can try comparing to your results?
 - You do not explain what made you choose a monolingual Swedish wav2vec model as your starting point. In intro you link to the wav2vec model finetuned by the National Library of Norway. That is as far as I can tell based on a multi-lingual wav2vec model instead of a monolingual Swedish model. Also, please add a HuggingFace link for the Swedish wav2vec model in a footnote, because they have several and it is not obvious which one you use.
- I find the expression "Recognition was performed in two different modalities" confusing and unconventional (modalities usually refer to raw inputs not LMs or other model components/knowledge) when the more conventional way in my opinion would be to write something like "We test decoding both with and without external language model..."
- "... it makes sense to inspect the NPSC and Rundkast test sets separately..." Could you also reflect this in Table 2 i.e. order the columns so that in-domain test sets come first and are next to each other followed by the out-of-domain test sets?
- "The language model presumably reduces the number of spelling mistakes in all
models." Have you actually looked at the decoding errors? Your argument would be more convincing if you could talk about which errors decrease when decoding with an LM instead of just guessing.
- An uninitiated reader would benefit if you could explain (e.g. somewhere in section 3) that the finetuned models produce character level output (assuming that is the case). Your findings in the lexical analysis make more sense if the reader understands that.
- "We suspect that the difference in hapax count between the ground truth and the automatic transcriptions is due to dialect and pronunciation-based misspellings to a large extent." Could you actually compare the lists with e.g. diff or comm instead of making guesses?
- "Another source of errors, ... , is differences in the standardization in the different datasets." This is very confusing to read, because 2.5 implied that you did standardize all the data sets in the same way but here you write that there was still differences. Only in conclusions you clarify that these errors were found despite your original standardization efforts.

**Paper Type:**

Long paper

---

### Official Review · Reviewer_f3pC · 2023-03-12
**An intersting and thoroughly written paper on how to combine data sets with both formal and colloquial speech for improving ASR.**

**Rating:** 8
**Confidence:** 3

**Review:**

The paper is of high quality and is clearly written. The originality is in the application of the methodology to new combinations of data sets. The work is significant in that it covers a wide variety of language use.

The pros of the paper are a detailed error analysis and a discussion of what improvements are gained from the various data sets.

The cons of the paper are not worth mentioning.


**Paper Type:**

Long paper

---

### Decision · Program_Chairs · 2023-03-17

Accept